# Fine Structure of the Mouthparts of Three *Tomicus* Beetles Co-Infecting *Pinus yunnanensis* in Southwestern China with Some Functional Comments

**DOI:** 10.3390/insects14120933

**Published:** 2023-12-07

**Authors:** Yajie Cui, Mengdie Zhang, Haidi Zhu, Pei Yang, Bin Yang, Zongbo Li

**Affiliations:** 1Key Laboratory of Forest Disaster Warning and Control in Yunnan Province, Southwest Forestry University, Kunming 650224, China; cuiyajie7@outlook.com (Y.C.); mdzhang9@swfu.edu.cn (M.Z.); czhuhaidi@outlook.com (H.Z.); yangbin48053@swfu.edu.cn (B.Y.); 2College of Traditional Chinese Medicine, Yunnan University of Chinese Medicine, Kunming 650500, China; 521yangpei@163.com; 3Key Laboratory for Forest Resources Conservation and Utilization in the Southwest Mountains of China, Ministry of Education, Southwest Forestry University, Kunming 650224, China

**Keywords:** *Tomicus*, chewing mouthpart, mandibular shape, sensory organ, intraspecific variability

## Abstract

**Simple Summary:**

Three bark beetle species, *Tomicus yunnanensis*, *T. brevipilosus*, and *T. minor*, are the most economically significant pests of the Yunnan pine in Southwestern China. Chemical and physical communication play key roles in various life activities. In this study, we described the fine structure of the adult mouthparts of these three *Tomicus* species using scanning and transmission electron microscopy. We identified three types of mandibular shapes, which match with their biomechanical properties, their ability to process food, and their preferred foraging locations on tree trunks. Eleven types of sensilla were discernible, including sensilla basiconica, sensilla twig basiconica, sensilla coeloconica, sensilla chaetica, sensilla trichoidea, and sensilla digitiformia. The function roles of each sensilla type were given based on its distribution and structures, especially on the internal structures such as dendrites and tubular body. Most chemoreceptors occur on the palpal tips. No significant differences among the sexes or species were identified; however, intraspecific variability in the number of sensilla twig basiconica 3 and sensilla digitiformia sensilla was evident. These findings will aid future studies of the feeding niches and reproductive behaviors of *Tomicus* beetles.

**Abstract:**

*Tomicus yunnanensis*, *T. brevipilosus*, and *T. minor* are the most economically significant pests of *Pinus yunnanensis* in Southwestern China. Chemical and physical factors play critical roles in diverse biological activities. Here, we describe the fine structure of the adult mouthparts of these three *Tomicus* species using scanning and transmission electron microscopy. We identified three types of mandibular shapes, which determine their biomechanical properties, their ability to process food, and their preferred foraging locations on tree trunks. Eleven types of sensilla were discernible, including sensilla basiconica (Sb.1–2), sensilla twig basiconica (Stb.1–3), sensilla coeloconica (Sco), sensilla chaetica (Sch.1–2), sensilla trichoidea (Str.1–2), and sensilla digitiformia (Sdi). Each basiconic sensillum occurs on the palpal tips and is innervated by 2–6 dendrites. Sb.1 are gustatory receptors, Sb.2 are olfactory receptors, and the three other sensilla have dual taste and mechanical functions. Sco, Sch, and Str are mechanoreceptors. Sdi are mechanical vibration receptions, given that they are innervated by one dendrite with numerous dendritic branches into the nonporous cuticle. No significant differences among the sexes or species were identified; however, intraspecific variability in the number of Stb.3 and Sdi sensilla was evident. These results will aid future studies of *Tomicus* beetle behaviors.

## 1. Introduction

Insects have evolved diverse mouthparts (e.g., biting, sucking, piercing) to handle various organic materials [1]. When grinding and chewing solid food, mouthpart components that convert the food into fragments have frequently evolved in various hexapods, especially winged insects. These structures determine the biomechanical characteristics of mouthparts, reflect the adaptation of insects to specific diets, and optimize feeding performance [1,2,3,4]. Differences in the feeding apparatus can facilitate coexistence among co-occurring species, which can reduce interspecific competition and enhance the utilization of limited food resources [5,6,7]. Convergence in the size and morphology of the mandible associated with diet-induced head allometry has been observed in acridids and caterpillars, and phylogeny has no major effect on these patterns [2,6]. Plants provide a reliable and nutritious source of food. However, they produce defensive compounds, such as alkaloids, glycosides, and terpenoids, that prevent herbivory by insects [8]. The evolution of monophagy and oligophagy in insects is thus key for their ability to feed on plants; the chemical and physical characteristics of plants can be detected by insects on initial contact. After landing on the surface of plants and sampling plant fragments, sensory cues are detected by numerous sensilla located along the mouthparts [1,2,9,10]. Several researchers have studied the modifications and adaptive functions of insect mouthparts, especially their roles in feeding [1,3,9]. Many of these studies have focused on the sensilla and sensory mechanism of insect mouthparts, especially in megadiverse coleopteran beetles [11,12,13,14,15,16,17,18,19]. However, the form and function of the mouthparts of sympatric insects, such as forest insect pests, which use similar or limited foods and experience intense interspecific competition, remain unclear.

Three bark beetle species within the genus *Tomicus* (Coleoptera: Curculionidae: Scolytinae) co-occur in Southwestern China: *T. yunanensis* Kirkendall & Faccoli 2008, *T. brevipilosus* Eggers 1929, and *T. minor* Hartig 1834 [20]. In central Yunan Province, all three *Tomicus* beetles can co-infest *Pinus yunnanensis* and kill healthy trees; these three species have been responsible for ca. 25% of the death of Yunnan pine trees over the past 30 years [20,21,22]. Similar to *T. piniperda* [20], they need to feed on several young shoots and trees when they are sexually mature, and these mature adults locate suitable breeding trees, where mating occurs, via a series of cues, including host and nonhost odors, pheromones, courtship acoustics, and cuticular hydrocarbons (for details, see Figure 1 in the Materials and Methods). On trees already infested with *T. yunnanensis*, *T. brevipilosus*, and *T. minor*, beetles colonize their preferred part of the pine trunk to lay eggs to minimize intra- and interspecific competition. The larvae feed in the phloem and form larval galleries perpendicular to the egg galleries, which induces tree mortality due to water and nutrient deficiencies. In such situations, feeding niche overlap (e.g., vertical food gradients, timing, importance) and food utilization can further affect the population dynamics of the three sympatric *Tomicus* species. The evolution of differences in mouthpart morphology and sensory organs is also necessary for enhancing the utilization of limited food resources [2,7]. Overall, the life cycle of each *Tomicus* species reflects adaptation to boring and communicating within the host pines. Signaling from the host trees and conspecific mates, particularly olfactory, gustatory, and vibrational signals, can convey important information to bark beetles, which facilitates host selection, mate searching, and oviposition [10,23,24,25].

Like other insects, the sensory structures of bark beetles are mainly distributed on the antennae and mouthparts, which are specialized for sensory reception and food performance, respectively [1,9,26]. Their antennae and mouthparts also contain various types of sensilla, which sense stimuli associated with smell (olfactory sensilla), taste (gustatory sensilla), CO_2_ (CO_2_ sensitive sensilla), vibration (mechanosensitive sensilla), temperature (thermosensitive sensilla), and humidity (hygrosensitive sensilla) [12,15,18,19,23,25,27,28,29]. According to recent observations (Figure 1), the mating behavior of each *Tomicus* beetle begins when one male individual touches another using the antennae and mouthparts, which is similar to the mating behavior of the coffee berry borer, *Hypothenemus hampei* (Ferrari, 1867) [30]. Wang et al. (2012) [28] examined the antennal morphology and sensilla ultrastructure of *T. yunnanensis*, *T. minor*, and *T. brevipilosus* and found that some wall-pore sensilla have olfactory and gustatory functions. However, the form and structure of the components of the mouthparts of these three *Tomicus* beetles, especially morphological differences in the mandible and sensory palps, have not yet been studied. These research gaps have limited our understanding of the niche utilization patterns and reproductive behaviors of these beetles in Yunnan pine trees.

Here, we studied the form and structure of the mouthparts of *T. yunnanensis*, *T. minor*, and *T. brevipilosus* using scanning and transmission electron microscopy. In light of previous behavioral observations (Figure 1), we (i) characterized a variation in the general morphology and sensory organs among individuals, among the sexes, and among the three *Tomicus* species, (ii) determined whether mandible size and morphology are associated with niche utilization patterns in response to intra- and interspecific competition, and (iii) identified the sensory organs that sense chemical and physical signals from host trees and conspecific mates. The results of this study provide valuable information that will aid future studies of the feeding and reproductive behaviors of these beetles and forest-integrated pest management.

## 2. Materials and Methods

### 2.1. Species Identification and Their Bioecological Characteristics

*T. yunnanensis*, *T. brevipilosus*, and *T. minor* have similar external morphologies, but they can be easily distinguished by granules or punctures on the second interstria along the declivity and length of the elytral interstrial hairs and hair arising from punctures [31]. Female and male individuals can be distinguished by the shape (semicircular vs. rectangular) and size (big vs. small) of the last abdominal tergites; live beetles can be sexed by the chirps produced by the male stridulation apparatus [31]. Finally, live beetles of each sex of each species identified in this study were temporarily stored in vials in a fridge at 4 °C until processing for scanning and transmission electron microscopy.

At our study site, JiuLong Shan Forestry Station (25°0′35″ N, 103°7′15″ E), located in central Yunan, Southwestern China, *T. yunnanensis*, *T. brevipilosus*, and *T. minor* co-occur and occupy their preferred niches on the trunks of Yunnan pine (Figure 1). During the first life cycle phase, which is also called the shoot-boring mature phase, the newly emerged adults fly to the canopy of a nearby stand of young pine trees, where they tunnel and feed on the shoots and reach sexual maturation from May to the end of October. Positive taxis has been observed in young pine shoots in response to a few attractants, such as α-pinene and β-myrcene [20,32]. After maturation feeding in the shoots, the second life cycle phase, the trunk-boring reproductive phase, begins in mid-November. This is characterized by an extended flight period, and adult beetles have several sister broods [21]. During flight, beetles are guided to dying or stressed pine trees, which emit a blend of volatiles, including α-pinene, β-myrcene, 3- carene, and α-terpinolene [32,33]. All three *Tomicus* species arrive and aggregate on the crown, and the outermost shoots are colonized first. However, the arrival time in both sexes of *T. yunnanensis* is always earlier than that of both sexes of *T. minor* and *T. brevipilosus*. Beetles produce and release the same aggregation pheromones, such as *cis*-verbenol and *trans*-verbenol [22,24], to induce mass attacks and weaken the tree at the crown of the host pine. A few weeks later, these beetles progressively spread down to their preferred positions along the pine trunk, which depends on intra- and interspecific competition as well as feeding niches [5,22]. At this point, the females excavate the nuptial chamber in the phloem and attract local males via host odors and pheromones. One female enters the chamber outside if one male produces the appropriate chirps via their sound organs; if not, the female remains in the hole. After this precopulatory behavior, the male touches the female’s elytra and pronotum with the antennae and mouthparts; a distinct courtship behavior occurs before the male mounts the female [31]. In the copulatory phase, the male positions his body perpendicular to that of the female and inserts his aedeagus into the female’s genital cavity. Typically, each sex only mates one time. After copulation, the female re-enters the chamber and expands the cylindrical gallery for egg laying. *T. yunnanensis* and *T. brevipilosus* excavate a typical longitudinal gallery, but *T. minor* excavates a vertical gallery. Finally, few post-copulatory adults may re-fly to the crown of pine trees and enter the next cycle.

### 2.2. Scanning Electron Microscopy (SEM)

More than 30 adults (more than 15 individuals for each sex) of each species were observed using field-emission scanning electron microscopy. Their head and mouthparts were dissected and placed in a glass tube with 2% glutaraldehyde solution in 0.1 M phosphate buffer (PBS, pH 7.2–7.4). After ultrasonic cleaning and graded dehydration (ethanol and pure acetone), the specimens were critical point-dried (Quorum K850, Ashford, Kent, UK) and sputter-coated with a film of gold (Cressington 108, Chalk Hill, Watford, UK). The samples were visualized using a Zeiss field emission scanning electron microscope (Sigma300, Jena, Germany) at 5–7 kV.

### 2.3. Transmission Electron Microscopy (TEM)

In light of the similar distribution of mouthpart sensilla on the three *Tomicus* beetles, only the labial and maxillary palps of decapitated *T. yunnanensis* at −20 °C were removed in the cold fixative solution; they were then immediately placed into fresh 3% glutaraldehyde solution in 0.1 M PBS (pH 7.4) for 2 h at 4 °C. After post-fixation with 1% osmium tetroxide, the specimens, completely stained with uranyl acetate, were embedded in Epon 812 and sliced on a Leica ultramicrotome (EMUC7, Wetzlar, Germany) with diamond knives. The grids were routinely stained and examined under a Jeol transmission electron microscope (JEM-1400Plus, Tokyo, Japan) at 120 kV.

### 2.4. Geometric Morphometric Analyses of the Mandible

The mandibular morphology of the three *Tomicus* species was compared using geometric morphometric methods. Only the female mandibles were used because no differences were observed among the sexes (Appendix A). Thirty homogeneous SEM images (e.g., scale, position, and orientation) of each species were used to generate TPS files using TpsUtil v 1.76 software [34]. The landmarks of each mandible in each image were placed along the axis of mandibular movement and digitized using TpsDig2 v 2.29 software [34]. All landmarks (See *an example of T. minor* in Section 3.1) in a single TPS file were numbered in the same order to ensure consistency with the same scale factor for different images. Thirty landmarks were recorded on each mandible of the three *Tomcius* species. To eliminate statistical errors, additional statistical analyses such as Procrustes ANOVA and correlation analysis (tpsSmall) were conducted to determine whether individual variance in the three species was sufficiently low [35,36]. Next, the TPS files were converted into NTS files (tpsUtil), which were imported into MorphoJ v 2.0 software for morphometric comparisons [37]. The samples were Procrustes superimposed in MorphoJ to discriminate shape and size and generate a matrix of Procrustes coordinates in an Euclidean space that is tangential to the Procrustes shape space [38]. Differences in the shape and size of three *Tomicus* species were evaluated using Procrustes ANOVA and principal component analysis (PCA) [38,39,40]. Following the PCA, canonical variate analysis (CVA) was conducted to determine the trait-based morphospace distribution of the three *Tomicus* species [40]. Finally, scatter plots of variation and deformation grid images were produced using MorphoJ software.

### 2.5. Terminology and Data Analysis

Mouthpart components and their sensilla types were described following the terminology of Altner and Prillinger (1980) [41], Vega et al. (2017) [12], and Shi et al. (2021) [14]. The numbers of sensilla were determined on the basis of the dorsal and ventral sides of the mouthpart elements. The mean length and basal diameter of each sensillum type were measured using ImageJ v 1.53t software (National Institutes of Health, Bethesda, MD, USA) on at least 20 sensilla of each type. Photoshop CC 2019 (Adobe Systems, San Jose, CA, USA) was used to clean the background and determine the distribution of sensilla on the maxillary and labial palps of the three *Tomicus* species. Wheater & Evans’s method [42] was used to measure various dimensions of the mandible, including the length (mesal edge), basal width, and chord length (mandibular articulation to apex). Turkey’s multiple comparison test was conducted to detect the significance of differences among groups using R v 4.3.1 software (R Core Team, 2023). All values are mean ± standard deviation.

## 3. Results

### 3.1. General Form and Structure of the Mouthparts among Three Tomicus Beetles

The mouthparts of each adult *Tomicus* beetle were typical biting mouthparts containing one labrum, two unjointed mandibles, two maxillae, and one labium (Figure 2), which together formed a preoral cavity. In general, the morphology and components of the mouthparts were similar among sexes and species (Appendix A); however, slight differences in several sensilla on the apex of the palps (Figure 1 inset) and mandibular apical incisor (Figure 3) were observed. The labrum, equivalent to the upper lip, is located at the anterior margin of the epipharynx beneath the anterior median section of the epistome (Appendix A), and there are several sensilla trichodea 1 (Str.1) on its cuticle. Below the upper lip are a pair of mandibles, which are heavily sclerotized, and triangular structures with four incisors (apical, subapical, median, and molar) for grinding food; the articulating condyles and fossa are arranged in a trochlear pattern (Appendix A). The external view of the mandible shows two cuticular impressions in the dorsal and ventral views. The former has one sensilla chaetica 1 (Sch.1), and the latter has two. The maxillae are a pair of jointed elements for eating food with the lateral position of the head behind the mandible. Each maxilla consists of four segments: the cardo, stipes, galea–lacinia complex, and maxillary palp (Appendix A). The cardo is a stout, saddle-shaped basal section that articulates flexibly via a membrane connected to the head. It bears a few sensilla chaetica 2 (Sch.2). The stipes are movable, flat-like structures located near the cardo that are not a single subsegment. This structure also contains the same type of Sch.2, similar to the cardo on the outside surface. The galea–lacinia complex is a fused structure between the galea and lacinia, and a distinct separated line on the external lateral area is lacking. The maxillary palp and stipes are also fused with the subgalea on the external lateral site, but a separate line is present on the internal lateral area. The distal part of the lacinia is armed with teeth and spines used to cut or chew hard foods. These two fused structures bear four types of sensilla: sensilla trichodea 1 (Str.1) and 2 (Str.2) and sensilla chaetica 1 (Sch.1) and 2 (Sch.2). In general, the complete surface of each maxillary palp with three-jointed segments is exposed to the environment; the distal ends of the maxillary palp are flattened and have a high concentration of apical sensilla (Figure 1 and Appendix A). The labium, also called the lower lip, is a simple structure composed of the postmentum, prementum, paraglossa, glossa, and labial palpi (Appendix A). The postmentum articulates with a bifid prementum and surrounds the basal section of the labium; it is subcylindrical, with the anterior ventral area being strongly retuse. The postmentum and prementum have a scaled external cuticle with no sensilla but an internal cuticle with dozens of Str.1 and Sch.1–2. The paraglossa and glossa form a fused ligula structure. The unpaired ligula is situated between the labial palpi, is approximately heart-shaped, and occupies the largest portion of the dorsal view. The internal surface of the ligula contains some Str.1, Sch.1, and sensilla basiconica 1 (Sb.1). Attached to the prementum is a pair of three-jointed labial palpi. The tip ends of the first and second joint have a row of Sch.1, but the third joint has a flattened area with a protrusion of the apical sensilla; the labium appears to play a role in the manipulation of food during grinding.

### 3.2. Morphological Divisions of the Mandible among Three Tomicus Beetles

The various dimensions of the mandibles of the three *Tomcius* beetles are shown in Table 1. The mandibular mechanical advantage of adult *T. minor* is slightly larger than that of the other two *Tomicus* beetles; no significant differences were observed among beetle species (F_(2,27)_ = 0.639, *p* = 0.536). According to the geometric morphometric analysis, the first two principal component axes of the mandible explained 57.99% of the variance (Figure 3A). The thin-plate spline of the mandible was represented by two characteristic changes in the axis of mandibular movement and the curved sharp tip of the apical incisor (Figure 3B). The shape of the mandible of *T. yunnanensis* is rudimentary, whereas the opposite shapes were observed for *T. brevipilosus* and *T. minor* (i.e., contraction and expansion from the same site). The average shape of the apical incisor of the mandible of *T. brevipilousus* was straight, and the tip was pointed forward. There was a small decrease in the distance between the apical and subapical incisors. Their mandibular shapes were sharper. The average profile of the mandible of adult *T. minor* included a broad apical incisor that was pointed inward; the mandible of adult *T. minor* is stout. This result was consistent with the size of the 95% equal frequency ellipses (Figure 3A). According to the differences above from the PCA, a CVA revealed significant differences in mandible morphologies among the three *Tomicus* beetles (Figure 3B). Mahalanobis and Procrustes distances among the three beetles are shown in Table 2. The *p*-values obtained with 10,000 permutations for the Mahalanobis and Procrustes distances were all far less than 0.0001, confirming that these distances differed significantly.

### 3.3. Types, Distributions, and Structures of Mouthpart Sensilla

In all discernible taxa, six different types of sensilla were present on the cuticle of the mouthpart elements: sensilla basiconica (Sb), sensiilla twig basiconica (Stb), sensilla coeloconica (Sco), sensilla chaetica (Sch), sensilla trichoidea (Str), and sensilla digitiformia (Sd). According to their distinct morphological differences, Sb, Sch, and Str were subdivided into two subtypes, and Stb was subdivided into three subtypes (Table 3 and Figure 4 and Appendix A). Several cuticular pores are visible on the mouthpart elements (Appendix A). Sb2 and Stb (3 subtypes) were concentrated on the terminal tip of the maxillary and labial palps of each *Tomicus* species and were jointly regarded as apical sensilla (Figure 2). The cell bodies of these sensilla lie at the base of the palp. The trichogen and thormogen cells continued up to the second segment of the palp. The projections of the thecogen cells extend forward to the third segment, which surrounds the outer dendritic segments with a cuticular sheath (Figure 5A–D). Some tracheoles penetrate each segment of the maxillary and labial palps (Figure 5A–C,E), but no mouthpart sensilla are present. All these sensilla are regularly distributed in the three species (Appendix A).

#### 3.3.1. Sensilla Basiconica (Sb)

Sb.1 only occur on the ligula’s ventral sides on the labial palp of each beetle; they are mixed with long lacinial teeth and are significantly less than the length of the lacinial teeth (Appendix A). The single, slender, and straight cone has a smooth-walled shaft with 2–3 terminal pores at the blunt tip (Figure 4A and inset). Thick nonporous cuticular walls surround a small sensilla lymph cavity innervated by 2–3 outer dendritic branches (Figure 5F). The length of these thick-walled sensilla varies from 44.4 ± 2.7 µm in *T. minor* males to 50.8 ± 3.3 µm in *T. brevipilosus* males, and their number ranges from 35.9 ± 0.3 in *T. minor* males to 36.1 ± 0.4 in *T. yunnanensis* females (Table 4 and Table 5).

Sb.2 are the largest apical sensilla and only occur in the center of the tip of the palps. These sensilla are smaller but more robust and basiconic compared with Sb.1 (Figure 4A,B and Appendix A). They have a smooth cuticle; the sensillum wall is nonporous at most of the cone shaft, and many open circular pores are present in the terminal tip (Figure 4B and inset). Pore structures are visible in the section and perforate into the thin cuticular cavity with numerous dendritic branches (Figure 5(G1)). Shortly below the tip, the cuticle becomes gradually thicker and denser and surrounds the large lumen innervated by multiple dendritic branches (Figure 5(G2)). From the middle to the basal adjacent region, the dendritic pattern innervating into the lumen does not present any apparent changes. Below the unobvious socket, a thormogen cell within a well-developed apical microvilli border surrounds 4–5 sensory neurons (Figure 5C,E,G3,H). Some mitochondria are evident at the contact area (Figure 5E,G3).

#### 3.3.2. Sensilla Twig Basiconica (Stb)

Stb.1 are the second largest sensilla on the apical sensilla field and are located near Sb.2 (Table 4 and Appendix A). They are located in an unobvious socket and have a smooth shaft with a depressed tip (Figure 4C and inset). The cuticle of the tip has eight orderly arranged finger-like protrusions that form a terminal pore in the central joint. There were 16 individual Stb of similar size on the mouthparts of each *Tomicus* beetle. From the tip to the base, the sensillum lumen widens gradually and permanently houses a dense dendritic sheath. The lumen is encircled by a thick nonporous wall and is innervated by four unbranched dendrites (Figure 5(I1,I2)). One of them always terminates as a tubular body attached to a socket cuticle (Figure 5(I3)). An individual dendritic sheath always separates the single tubular body-forming dendrite from the others, which extends into the sensilla shaft (Figure 5(I3)).

Stb.2 are adjacent to the aforementioned apical sensilla, Sb.2 and Stb.1 (Table 3 and Figure 5S). In contrast to Stb.1, the ends of Stb.2 have a papillate protrusion that extends from the center of the depressed tip (Figure 4D). In the TEM section, a single terminal pore can be distinguished from the interspace between the protrusions and sensillum edge, where it merges in the central lumen of the sensillum (Figure 5(J1)). The following transverse section revealed a thin dendritic sheath without dendrites inside the narrow lumen (Figure 5(J2)). Further basally, the lumen gradually widens, and the dendritic sheath houses four unbranched external dendrites (Figure 5(J3)). Basally, up to six sensory neurons innervate the Stb.2 lumen, but a separated dendrite may contain a tubular body at the socket (Figure 5(J4)).

Stb.3 are the smallest sensilla among the apical sensilla (Table 4 and Appendix A). These sensilla are situated at the edges of the tip of the palps and are arranged in a ring pattern. The basiconica-like Stb.3 possess a two-layered, robust cylindrical cone nested within a spherical apex with a smooth cuticle and a conspicuous cuticular pore (Figure 4E). The terminal pore penetrates the central lumen of the shaft, where a thin dendritic sheath is present (Figure 5L,L1). Following the pore, membranous structures gradually become large inside the sheath, and the shaft cuticle is thick and nonporous (Figure 5K,L,L2–L4). Basally, four unbranched dendrites innervate the lumen of Stb.3, but one of them disappears in the proximal segment of the spherical apex (Figure 5K,L,L5). This indicates that the separated dendrite may contain a mechanosensory tubular body at the socket.

#### 3.3.3. Sensilla Coeloconica (Sco)

Sco are exclusively located on the lateral side of the third subsegment of the maxillary palp (Figure 5A,C and Appendix A). It is visible as a small dome ventral to the sensilla digitiformia (Figure 4F). Externally, it has a conspicuous lateral molting pore that is lacking in the Sco of other insects. The lumen of the sensilla is encircled by a thick nonporous cuticle and is innervated by one unbranched dendrite ending in a tubular body (Figure 5M). The dendrite is tightly enclosed by a dendritic sheath attached to the cuticle below the socket.

#### 3.3.4. Sensilla Chaetica (Sch)

Sch.1 are the only type of sensilla that are common in all the mouthpart elements but the labrum (Figure 4G and Appendix A), and they vary dramatically in length. The shortest is ca. 8.3 µm in length, and it occurs on the lateral side of the labial palp; the largest one has a length of 161.3 µm and is located in the maxillary palp. The cuticle of the sensillum has a thick, smooth, and nonporous wall, which is not innervated by dendrites, according to the TEM image (Figure 5N).

Sch.2 are mainly distributed on each element of the maxillae and labium (Figure 4H and Appendix A). This sensillum is characterized by a raw tooth tip formed by a firm spine protruding from the edge of the shaft. These teeth are usually in one line along the shaft of the sensillum, but occasionally two lines are observed (Appendix A). Like Sch.1, the length of Sch.2 ranges from 6.5 to 198.3 µm on different mouthpart elements in each species. No dendrites are innervated in the thick and nonporous lumen (Figure 5O).

#### 3.3.5. Sensilla Trichoidea (St)

St.1 are straight or slightly curved in the middle with sharp distal tips and flexible sockets (Figure 4I). They are situated on all mouthpart components except the mandible (Appendix A). This sensillum varies from 7.3 to 32.4 µm in length. Thick nonporous cuticles surround a small inner lumen that is not innervated by dendrites (Figure 5P).

St.2 are similar to St.1 but are large. They possess blunt tips with no detectable depressions and flexible sockets (Figure 4G). These sensilla are concentrated on the lateral field of the galea–lacinia complex. Their length ranges from 14 to 27 µm in each species. The cuticle of St.2 exhibits a thick and nonporous wall that is not innervated by dendrites (Figure 5Q).

#### 3.3.6. Sensilla Digitifomia

Sdi are exclusively located on the lateral side of the third subsegment of the maxillary palp. They are chip- or finger-shaped and lay flat in a long oval cuticular depression (Figure 4K and inset and Appendix A). Their tips are blunt, occasionally bifurcated, and always point toward the terminal end of the maxillary palp. The externally visible cuticular shaft is ca. 12 µm and lentil-shaped in the transverse section (Figure 5(R1,R2)); it is 2 µm wide and maximally 3 µm thick. A channel, approximately circular in the transverse section and just over 0.7 µm in diameter, runs along the whole shaft. Numerous branched dendrites innervate the lumen of the chip, but the dendritic profiles do not show a lamellar arrangement. Their number is reduced but their diameter increases towards the base of the chip shaft. Finally, only one ensheathed outer dendrite is present at the bottom of the socket. All profiles in the chip shaft are branched from this single dendrite. Otherwise, the structure of the integument of the cuticular depression and the exocuticle is homogeneous, indicating that the former has no additional sensory functions.

#### 3.3.7. Cuticular Pore (Cp)

The Cps are found on the external side of the mandible and each side of the maxillae and labium of each species (Appendix A). A single Cp comprises a small oval hole with a long diameter of 0.77 ± 0.17 µm and short diameter of 0.64 ± 0.13 µm. A cuticular apparatus in the form of an irregular web, or regular, fine, transversely orientated lamellae, is usually present in the bottom of the hole (Figure 4L). However, the ultrastructure of this apparatus remains unclear.

### 3.4. Intra- and Interspecific Sensilla Variations among Three Tomicus Beetles

All six types of sensilla, including different subtypes, occur on the mouthparts of each individual and sex in the three *Tomicus* beetles (Figure 1 and Appendix A). There were no statistically significant differences in the length (Table 4) and number (Table 5) of each type of sensilla among sexes and species. The distribution of sensilla on each mouthpart element was similar. However, a few discernible differences in Stb.3 and Sdi on the last subsegment of the maxillary and labial palps were observed on individuals of each *Tomicus* species (Figure 1 and Figure 6A,B). The number of Stb.3 or Sdi on both sides was asymmetrical. For example, the number of Stb.3 on the left or right maxillary (labial) palp varies from 9 to 14 (13); higher density can only be achieved by increasing the amount per unit of volume. Most of the palps have 11 Stb.3, accounting for 32.8% and 31.2% of all sampled maxillary and labial palps, respectively; palps with 12 (10) Stb.3 accounted for 18.8% (25.0%) and 18.8% (18.8%) of all the maxillary and labial palps sampled, respectively (Figure 6C,D). The number of Sdi on each palp ranges from 4 to 6; most palps have five Sdi, which accounts for 64.7% of the total amount. Maxillary palps with four Sdi are rarely observed and only comprise 5.9% of all palps (Figure 6E). Numbers of Stb.3 and Sdi on each side of the maxillary and labial palps usually only differ by 1 to 3 (Figure 1).

## 4. Discussion

The structure and configuration of mouthparts in Coleoptera are conserved and plesiomorphic; the chewing-type mouthparts contain five main components: labrum, mandible, maxillae, labium, and hypopharynx [1]. Each mouthpart element works with others to handle food; for example, the mandible and maxillae are used to manipulate food. However, distinct morphological modifications are observed in immature and adult beetles, and these are associated with differences in feeding habits; some examples of these modifications include clypeolabral fusion in Hydrophilidae larvae [43], falcate mandibles without mola in Staphylinidae adults [44], the galea–lacinia complex in Scolytinae adults [12], the small setose lacinia in Scarabaeidae adults [17], and the increased movability of the maxillary and labial palps in Chrysomelidae [45]. In this study, we described the mouthparts of three *Tomicus* beetles, including differences in the mandible and various types of mouthpart sensilla. Compared with those of other scolytid beetles, including *Dendroctonus* spp. [46], *Ips acuminatus* [15], *I. typographus* [14,18,47], *Ips subelongatus* [47,48], *Hypothenemus hampei* [12], *Platypus cylindrus* [49], *P. koryoensis* [16], and *Euplatypus parallelus* [11], the mouthparts of the three *Tomicus* species appear to be adapted to inhabiting the inner bark environment; however, some variation in the mouthparts might reflect variation in the parts of the trunks used by *Tomicus* species.

### 4.1. General Characters and Modification of Mouthpart Elements in Scolytid Beetles

Scolytid beetles, including bark and ambrosia beetles, are highly adapted to life within trees and thus possess mouthparts that are capable of excavating tunnels to facilitate the acquisition of solid food or the transport of symbiotic fungi [14,15,48]. The mouthparts of the three *Tomicus* species were similar to the chewing-type mouthparts of most phytophagous coleopteran beetles, which are used to make tunnels in tree shoots and stems. The mouthparts of these *Tomcius* species are composed of the labrum and three other appendages: mandibles, maxillae, and labium. The labrum, an obsolete triangular element, is common in all scolytids and might be represented by a residual part of the epipharynx’s anterior margin beneath the epistoma’s median section. The mandible, maxillae, and labium constitute the moveable mouthpart together with the maxillary and labial palp, which can be used to grasp and masticate plant substrates, manipulate them in the preoral space, and direct them to the mouth opening [1,15]. Similarly, modifications in one or more elements may be associated with feeding preferences; Ref. [15] showed that the maxillary structure of the mouthparts can serve as mechanical carriers of pathogenic microorganisms. Furthermore, apical sensilla on the terminal ends of the mouthpart palps are regarded as olfactory and gustatory receptors that can aid decision making as well as communication with hosts and mates (Figure 1). For example, *I. paraconfusus* Lanier, 1970 males cease their boring activity only when they encounter the outer bark of the nonhost white fir *Abies concolor* (Gordon), which indicates that a decision had been made following gustatory verification [10]. All traits can be considered functional [50]. However, the functional significance of the scaly cuticular structures on the external view of the prementum on the three *Tomicus* beetles (Appendix A) remains unclear. One possible explanation for the functions of these scaly structures is that they facilitate the transfer of food particles to the mouth. Indeed, these differences can be used to distinguish among these *Tomicus* species.

The heavily sclerotized mandibles, referred to as the first jaws, are typical biomechanical tools with blade-like incisors that cut food particles via bite forces generated by an apodeme and a mandibular joint. The evolution of these structures indicates that only one simple change in morphology can greatly enhance the mechanical advantage and facilitate the utilization of unoccupied niches [3,6,42]. A detailed comparative study of the mandibles may reveal key specific characters; geometric morphometric analyses of the mandibles of the three *Tomicus* species provided important insights (Figure 3). We found a positive correlation between the incisor length and chord length, and no significant differences in the mechanical advantage were observed among the three species. However, some minor changes, such as contraction and expansion from the incisors and mandible joints (Figure 3B), require increased attention. For example, *T. minor*, which bore in the basal stem, tends to have stouter mandibles than *T. yunnanensis* and *T. brevipilosus*, which bore in the middle to the upper part of the stem. However, *T. brevipilosus* tends to have a sharper mandible than *T. yunnanensis* and *T. minor*. A positive correlation between head width behind the compound eyes and mandibular adductor muscles has been observed in a previous study (y = 0.329x + 0.257) [42]. The head width of the three *Tomicus* species ranged from 0.92 mm to 1.27 mm (*p* = 0.428). Whether mandibles are stout or sharp, minor modifications can enhance their use and bite force [2,3]. Field observations confirm these possibilities in Southwestern China (Figure 1). *T. yunnanensis* predominately colonizes the upper part of the pine stem, which is covered in smooth bark that can be easily cut, even if the mandibles have a basic shape with unmodified incisors and mandibular joints; however, *T. minor* and *T. brevipilosus* mainly colonize the lower stem, which has thick and corky bark, and tunneling requires powerful mandibles. Stouter and sharper mandibles can also facilitate the exploitation of more feeding niches; *T. brivipilosus* colonizes all parts of the trunk of *P. kesiya* Royle ex Gordon if the two other *Tomicus* species are absent [22]. *T. minor* also does the same on *P. yunnanensis*. However, their egg gallery ends deeper into the xylem than any other *Tomicus* beetles, which suggests that they possess stouter mandibles [20]. More geometric morphometric comparisons and 3D reconstructions will be needed to quantitatively characterize morphological modifications in scolytid beetles.

### 4.2. Putative Function of Mouthpart Sensilla

Many studies have reported that sensilla basiconica (Sb) are widely distributed on the antennae, maxillary palps, and labial palps in coleopteran beetles, including larvae [47,51,52] and adults [11,13,14,15,16,18,30,44,48,49]. Unlike porous wall Sb on the antennae, these sensilla have only some pores on the terminal tips, which are characteristic of Sb on the mouthparts. In general, the porous sensilla are typical chemoreceptors because the open pores allow odor molecules to enter the sensilla lymph and activate neuronal signals via odorant-binding and chemosensory proteins [18,53,54]. The external morphology of Sb.1 was similar to that of terminal pore sensilla (TP) [18] and sensilla basiconica type I (BA^1^) [14] in *I. typographus*. The positions of the two sensilla differ (Appendix A). Sb.1 are distinctly lower than the long and stiff lacinial teeth on the labium. In light of the 2–3 dendrites innervated into the lymph cavity, Sb.1 might act as gustatory receptors that sense the chemical properties of food during ingestion [9,18,41,49]. Sensilla with similar structures, locations, and numbers as Sb.2 are also referred to as single-walled sensilla (SW) [18,48] and sensilla styloconica subtype 2 (BA^2^) [14,51]. The multiple pores on the tips and 5–6 dendrites without tubular bodies are typical of olfactory receptors, and the terminally located pores serve as taste receptors [18,41,47,48,54]. Given that Sb.2 are the first sensilla that come into contact with the substrates, a dual taste/olfactory function is helpful for odor sensation and discrimination.

Sensilla with a fine morphology similar to sensilla twig basiconica (Stb) in this paper have been reported in the aforementioned bark and ambrosia beetles [11,12,14,15,16,18,19,47,48,49], *Cryptorhynchus lapathi* [55], rove beetles of the Staphylinine group [44], *Anoplophora glabripennis* [52], and *Leptinotarsa decemlineata* [56]. These Stb are also more prevalent on the terminal tips of the maxillary and labial palps. However, the Stb on the palps are shorter than the Stb on the antennae, and the size of the cuticular pegs differs in these sensilla [14,28,29,48]. Stb are highly exposed on the basis of their distribution on the mouthparts. These Stb terminate at approximately the same level as the mechanosensory bristles, including sensilla chaetica and sensilla trichodea. According to their positions and morphologies, Stb.1 are similar to TP in *I. typographus* [18] and *D. ponderosae* [19], sensilla styloconica subtype 1 (ST^1^) in *I. typographus* [14], sensilla twig basiconica III (T.b.3) in *I. subelongatus* [47], and sensilla twig basiconica subtype 5 (S.tb.5) in *C. lapathi* [55]. Stb.2 share morphological characteristics with the mechanosensory pegs in *I. typographus* [18], ST^2^ in *I. typographus* [14], S.tb.3 in *C. lapathi* [55], and uniporous sensilla styloconica in *Noctua pronuba* [57]; the Stb.3 correspond to those of ST^3^ in *I. typographus* [14], mechanosensory nipples in *I. typographus* [18], campaniform organs in *D. ponderosae* [19], and S.tb.6 in *C. lapathi* [55]. Compared with the antennal Stb, which have two, five, or six dendrites [18,28,41], the numbers of dendrites of Stb 1–3 are less variable (4–6) (Table 3). In light of the correlations in the structure and function of insect sensilla, the Stb in this study have a combined gustatory and mechanoreceptive function [18,19,41], given that they had four or six dendrites, with one terminating as a tubular body in the lymph lumen and at least one pore in the terminal tip. Dimensional differences in the Stb and Sb.2 can be observed on the palpal tips, which come into solid contact with the substrate, during the discrimination process; these sensilla thus simultaneously sense mechanical and chemical stimuli from the outer bark and the body of their mates.

Two sensilla coeloconica (Sco) near the tip of the palps of each *Tomicus* beetle had similar positions and morphologies in the campaniform organs in *D. ponderosae* [19], sensilla coeloconica in *Megabruchidius dorsalis* [58], and dome-shaped sensilla in *Pterostichus oblongopunctatus* [59]. However, the occurrence of Sco on the mouthparts is rare compared with that on the antennae in Coleoptera [14,48,58,60,61]. All Sco in this study have a lateral molting pore, which is usually lacking on the dome in other beetles. These might be mechanosensitive given that a dendrite with a tubular body is present [19]. However, these sensilla might function as combined thermoreceptors and chemoreceptors [41,59]; additional immunocytochemistry analyses and electrophysiological data of the functional neurons from the Sco in the palpal tips of *Tomicus* beetles are necessary [54,59].

Sensilla chaetica (Sch) and sensilla trichodea (St) are the most common types of sensilla on the mouthpart elements, with the exception of the palpal tips. The locations and morphologies of these sensilla are similar to those of other coleopteran species described in previous studies [1,11,13,14,15,16,18,19,28,29,46,48,49,58]. The dendrites do not innervate these sensilla and possess a basal tubular body, which is supported by the observation that dendrites are absent on the lymph cavity of the sensilla (Figure 5N–Q). Thus, Sch.1, which are prominent in the external view of the mandible, might respond to mechanical stimuli and help modulate the power output of the adductor muscles [9]. Sch.2 are highly similar to sensilla chaetica type I (CH^1^) or II (CH^2^) in *I. typographus* [14] and sensilla furcatea on the antennae of *Tomicus* species [28]; they probably sense mechanical stimuli from the substrates. The morphologies of St.1 on the mouthparts are similar to those of T.r.1 in *I. subelongatus* [48] and sensilla trichodea III in *D. ponderosae* [19]. Based on their location and morphology, St.1 might have mechanosensory functions similar to Sch.1 during contact. St.2 are only densely distributed on the lateral sides of the galea–lacinia complex and have a nonporous wall; they might be mechanoreceptors that help *Tomicus* beetles detect and prevent food from slipping back out of the mouthparts [9,14,41].

Sensilla digitiformia (Sdi) are frequently present on the palpal tips of Coleoptera, including immature and adult individuals [11,13,18,49], but they are never on the antennae [28,29,47,48,52,55]. These Sdi have been described as poreless sensilla in *I. typographus* [18] and sensilla placodea in *I. subelongatus* [47]. The smooth cuticle of the Sdi in *Tomicus* beetles is traversed by one dendrite with numerous branches innervated into the lymph cavity, which is typical for vibration-sensitive receptors [62], but not for thermosensitive and CO_2_ receptors [18,63]. In each *Tomicus* species, copulatory interactions can be triggered after the males have produced acoustic cues (Figure 1). However, no sound receptors have been identified in scolytid beetles [23]. Previous studies of insect acoustic communication indicate that Sdi might detect vibrational signals in bark beetle systems.

### 4.3. Intra- and Interspecific Variability in the Mouthpart Sensilla

The sizes, structures, and distributions of sensilla of all types are similar in the adults of both sexes of these three beetles, which is consistent with the results of previous studies of scolytid beetles [11,14,18,19,48]. For example, aporous mechanoreceptors such as Sch and St are randomly scattered over the cuticle of the mouthpart elements aside from the palpal tips; the porous chemoreceptors involved in Sb.2 and Stb.1–3 are densely arranged on the terminal ends of the maxillary and labial palps. However, some differences have been observed among scolytid beetles. For example, in *I. typographus* and *I. subelongatus*, the maxillary and labial palps contain 10–21 and 12–21 chemoreceptors [14,48]; *D. ponderosae* [19], *I. typographus* [14,18], *I. subelongatus* [48], *I. acuminatus* [15], *P. koryoensis* [16], and *E, parallelus* [11] have 6, 4, 5, 5, 5, 32, and 78 chemoreceptors in the lateral view of the maxillary palps, respectively. Furthermore, the total number of sensilla is greater on the tips of the maxillary palps than on the tips of the labial palps; thus, the tips of the maxillary palps are capable of detecting more environmental cues than the tips of the labial palps. Such individual variability in two palpal-tip sensilla, including Stb.3 and Sdi (Figure 6), to our knowledge, has not been previously reported in other Coleoptera. There might be an adaptive explanation for this observation, such as variation in the environmental stimuli experienced by different subpopulations. Individuals can evolve a wide range of adaptive traits when the same genotype produces different phonotypes under different environmental conditions, such as physical and chemical signals [4]. This might explain the variation in the mouthpart sensilla among *Tomicus* individuals; however, additional studies are needed to confirm this.

## 5. Conclusions

Using SEM and TEM, we described the fine structures of the mouthparts of *T. yunnanensis*, *T. brevipilosus*, and *T. minor*, which feed on different portions of the stem of *P. yunnanensis* to minimize competition. The mouthparts of each *Tomicus* species are conserved and plesiomorphic; the chewing-type mouthparts consist of the labrum and three other components: mandibles, maxillae, and labium. The shape of the mandible is rudimentary in *T. yunnanensis*, sharp in *T. brevipilosus*, and stout in *T. minor*, and this variation is associated with differences in the feeding locations of these species on *P. yunnanensis*. Six types of sensilla, including sensilla basiconica (Sb.1–2), sensilla twig basiconica (Stb.1–3), sensilla coeloconica (Sco), sensilla chaetica (Sch.1–2), sensilla trichoidea (Str.1–2), and sensilla digitiformia (Sdi), were identified. The pegs of Sb.1 at the tip have a few pores and are innervated by 2–3 dendrites, which suggests that they have gustatory functions. Sb.2 are olfactory receptors because each has multiple pores at the terminal tip and sends a highly branched dendrite into the lymph of sensilla. Three subtypes of Stb have dual gustatory and mechanical functions, given that each has a terminal pore and four or six dendrites with one terminating as a tubular body in the base of the sensilla. Sco are possibly mechanosensitive or combined thermo- and chemoreceptors. All types of Sch and Str have thick and nonporous cuticles that are not innervated by dendritic branches, suggesting that they have mechanosensitive functions. The Sdi are innervated by one dendrite, which produces numerous dendritic branches into a narrow channel within a chip-shaped protrusion of the nonporous cuticle; thus, digitiform sensilla are most likely mechanical vibration receptors. No significant differences among the sexes or species were identified. Substantial intraspecific variation was observed in the number of Stb.3 and Sdi on the right and left palps of three *Tomicus* species. These findings will aid future studies of the feeding niches and reproductive behaviors of *Tomicus* beetles.

## Figures and Tables

**Figure 1 insects-14-00933-f001:**
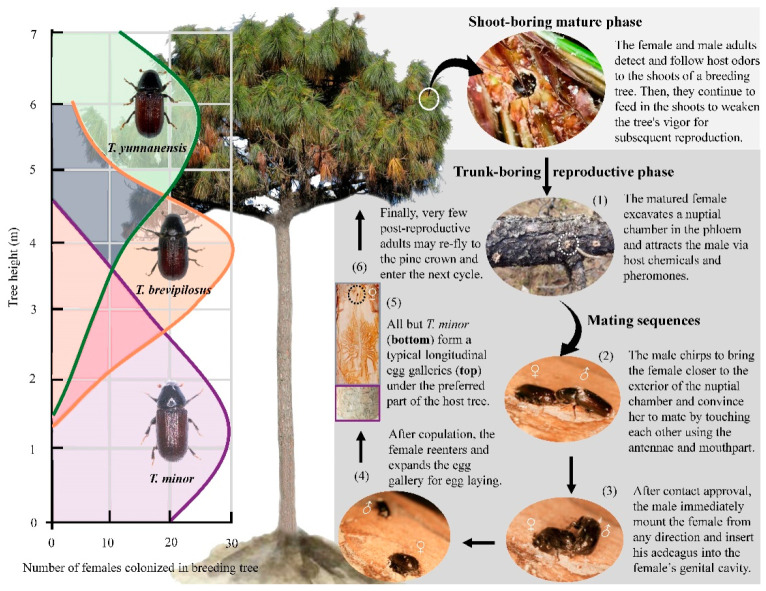
Niche utilization patterns (**left**) and mating sequences (**right**) among three *Tomicus* beetles on *P. yunnanensis*. Green (*T. yunnanensis*), orange (*T. brevipilosus*), purple (*T. minor*), and mixed color zones indicate (their) population densities at different heights on the trunk. Curved arrows show the feeding tunnel (solid circle) or the nuptial chamber (dotted circle). Heavy arrows with numbers indicate the behavioral sequence in a typical reproductive phase.

**Figure 2 insects-14-00933-f002:**
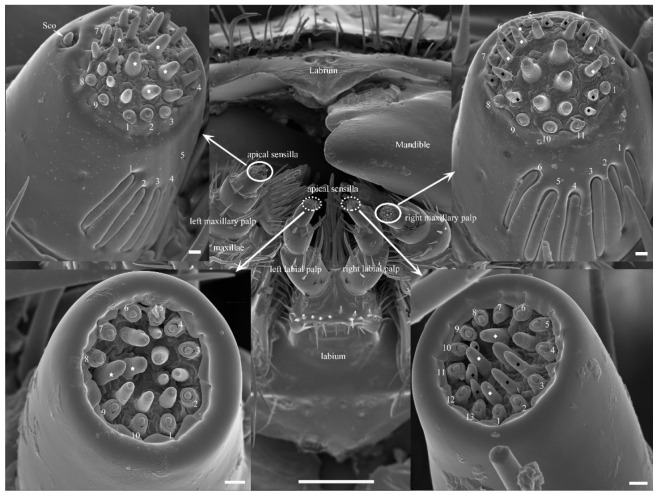
Mouthparts and their sensilla of a female *T. yunnanensis*. The solid and dotted circles show differences in apical sensilla between the left and right palps within an individual, respectively. White stars show sensilla basiconica 2; white and black diamonds show sensilla twig basiconica 1 and 2, respectively; the labeled numbers indicate the number of sensilla twig basiconica 3 and sensilla digitiformia on each palp within an individual. Sco, sensilla coeloconica. Scale bar = 100 µm, but all insets = 2 µm.

**Figure 3 insects-14-00933-f003:**
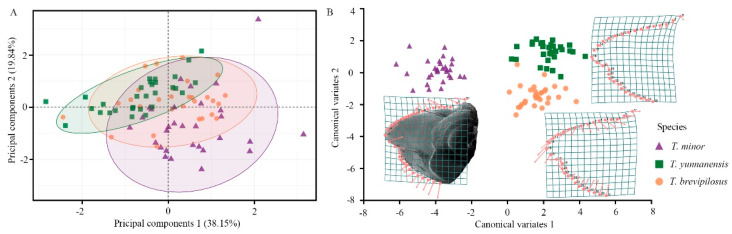
Geometric morphometric analysis of the mandible of three *Tomicus* beetles on the basis of principal component analysis (**A**) and canonical variate analysis (**B**). The equal frequency ellipses represent 95% confidence intervals. The thin − plate spines indicate the average shape of the mandible for each *Tomicus* beetle, which corresponds to the deformation grid of the recorded landmarks along the axis of mandibular movement.

**Figure 4 insects-14-00933-f004:**
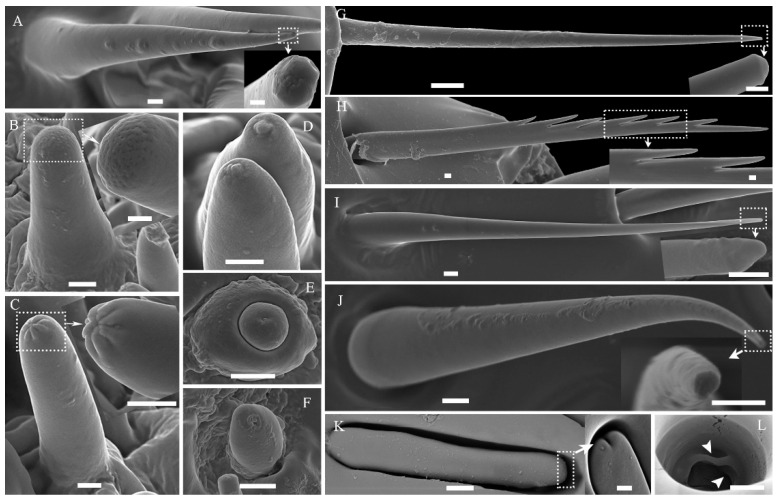
Fine structure of different types of mouthpart sensilla in three *Tomicus* beetles. (**A**) Sensilla basiconica 1 (Sb.1); (**B**) sensilla basiconica 2 (Sb.2); (**C**) sensilla twig basiconica 1 (Stb.1); (**D**) sensilla twig basiconica 2 (Stb.2); (**E**) sensilla twig basiconica 3 (Stb.3); (**F**) sensilla coeloconica (Sco); (**G**) sensilla chaetica 1 (Sch.1); (**H**) sensilla chaetica 2 (Sch.2); (**I**) sensilla trichodea 1 (Str.1); (**J**) sensilla trichodea (Str.2); (**K**) sensilla digitiformia (Sdi); (**L**) cuticular pore (Cp). The dotted box shows enlarged characteristics in the inserted picture. The white arrowhead shows transversely orientated lamellae. Scale bar, A–K = 1 µm; L and all inset = 500 nm.

**Figure 5 insects-14-00933-f005:**
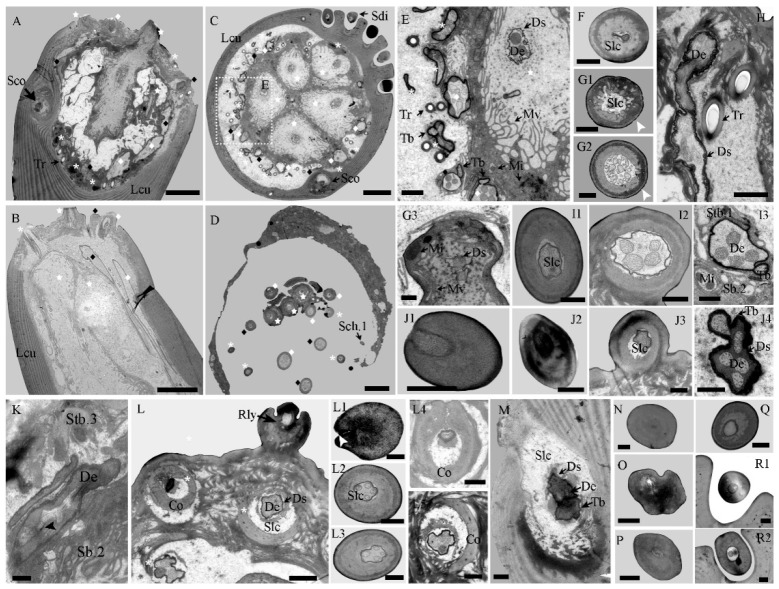
The internal structure of different types of mouthpart sensilla in *T. yunnanensis*. (**A**) Longitudinal section of the maxillary palp; (**B**) longitudinal section of the labial palp; (**C**) transverse section of the maxillary palp; (**D**) transverse section of the labial palp; (**E**) an enlarged area of the dotted box in C; (**F**) longitudinal section of sensilla basiconica 1 (Sb.1); (**G**) transverse sections of sensilla basiconica 2 from the tip (**G1**,**G2**) to the base (**G3**) in the cavity of the maxillary palp. (**H**) Longitudinal sections of sensilla basiconica 2 (Sb.2); (**I**) transverse sections of sensilla twig basiconica 1 from the tip (**I1**) to the base (**I2**,**I3**) in the cavity of the maxillary palp; (**J**) transverse sections of sensilla twig basiconica 2 (Stb.2) from the tip (**J1**,**J2**) to the base (**J3**,**J4**) in the cavity of the maxillary palp. (**K**) Longitudinal section of sensilla twig basiconica (Stb.3); (**L**) transverse section of Stb.3 from the tip (**L1**–**L3**) to the base (**L4**,**L5**) in the cavity of the maxillary palp; (**M**) transverse section of sensilla coeloconica (Sco); (**N**) transverse section of sensilla chaetica 1 (Sch.1); (**O**) transverse section of sensilla chaetica 2 (Sch.2); (**P**) transverse section of sensilla trichodea 1 (Str.1); (**Q**) transverse section of sensilla trichodea 2 (Str.2); (**R**) transverse section of sensilla digitiformia (Sdi) at the tip (**R1**) and middle (**R2**). White stars show Sb.2; white and black diamonds show Stb.1 and Stb.2, respectively; asterisks show Stb.3. The equilateral arrowhead shows a tiny pore on the sensilla wall. The black arrowhead shows the outer dendritic segment. De, dendritic; Ds, dendritic sheath; Iw, inner wall; Lcu, lamellated cuticle; Mi, mitochondria; Mv, microvilli; Slc, sensilla lymph cavity; Tb, tubular body; Tr, tracheole. Scale bar, (**A**–**D**) = 5 µm; (**E**,**H**,**L**) = 1 µm; (**F**,**G1**–**G3**,**I**–**K**,**L1–l5**,**M**–**R**) = 500 nm.

**Figure 6 insects-14-00933-f006:**
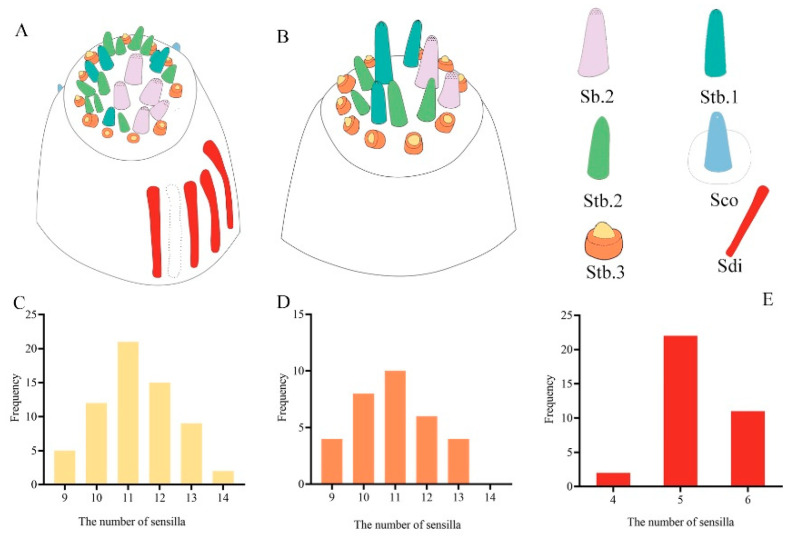
Intraspecific variation in the apical sensilla of the maxillary and labial palps in the three *Tomicus* beetles. (**A**) Maxillary palp; (**B**) labial palp; (**C**) numbers of Stb.3 on the maxillary palp; (**D**) numbers of Stb.3 on the labial palp; (**E**) numbers of sensilla digitiformia on the maxillary palp. The profile marked by the dotted lines shows a high frequency of sensilla at this site.

**Table 1 insects-14-00933-t001:** Dimension of the mandible in three *Tomicus* beetles.

Species	Length (μm)	Base Width (μm)	Chord Length (μm)	Mechanical Advantage *
*T. yunnanensis*	250.7 ± 31.8	170.9 ± 9.1	327.2 ± 16.4	0.52 ± 0.03
*T. brevipilosus*	241.6 ± 30.2	155.3 ± 13.4	292.8 ± 30.6	0.53 ± 0.05
*T. minor*	258.3 ± 11.8	166.5 ± 8.7	305.8 ± 21.0	0.55 ± 0.06

* The value is a ratio between base width and chord length.

**Table 2 insects-14-00933-t002:** Canonical variance analysis (CVA) of three *Tomicus* beetles.

Sort	Ty vs. Tb	Ty vs. Tm	Tb vs. Tm
Mahalanobis distances	4.295	7.279	6.788
*p* value for Mahalanobis distances	<0.0001	<0.0001	<0.0001
Procrustes distances	0.034	0.068	0.041
*p* value for Procrustes distances	<0.0001	<0.0001	<0.0001

Abbreviation of three *Tomcius* beetle: Ty, *T. yunnanensis*, Tb, *T. brevipilosus*, Tm, *T. minor.*

**Table 3 insects-14-00933-t003:** Morphological characteristics, distribution, and dendritic structure of the mouthpart sensilla in three *Tomicus* beetles.

Sensilla	Shape	Surface	Pores	Tip	Socket	Location	Dendrites
Sb.1	basiconic, straight, large, slender	smooth	subterminal pore	blunt	inflexible	Lig	2–3
Sb.2	basiconic, straight, small, robust	smooth	terminal pore	blunt	unobvious	Mpt, Lpt	5–6
Stb.1	basiconic, straight, small, robust	smooth	terminal pore	blunt	unobvious	Mpt, Lpt	4
Stb.2	basiconic, straight, papillate protrusion	smooth	terminal pore	blunt	unobvious	Mpt, Lpt	6
Stb.3	basiconic, straight, spherical apex	smooth	terminal pore	flat	unobvious	Mpt, Lpt	4
Sco	coeloconic, straight, sunken	smooth	lateral molting pore	blunt	flexible	Mpl	1
Sch.1	chaetic, straight	smooth	aporous	blunt	flexible	Md, Mx, Li	-
Sch.2	chaetic, straight, or slightly curved	saw-toothed	aporous	sharp	flexible	Mx, Li	-
Str.1	trichoid, straight, or slightly curved	smooth	aporous	sharp	flexible	Mx, Lr, Li	-
Str.2	trichoid, straight, or slightly curved	smooth	aporous	blunt	flexible	Glc	-
Sdi	digitiform in longish cuticular recess	smooth	aporous	blunt	flexible	Mpl	1

Sb.1 and Sb.2, sensilla basiconica 1–2; Stb.1, Stb.2 and Stb.3, sensilla twig basiconica 1–3; Sco, sensilla coeloconica; Sch.1 and Sch.2, sensilla chaetica 1–2; Str.1 and Str.2, sensilla trichoidea 1–2; Sdi, sensilla digitiformia; Glc, galeo–lacinial complex; Li, labium; Lig, ligula; Lpt, terminal segment of labial palp; Lr, labrum; Md, mandible; Mpl, lateral view of maxillary palp; Mpt, terminal segment of maxillary palp; Mx, maxillae. All abbreviations are consistent with the following tables and figures.

**Table 4 insects-14-00933-t004:** The mean length of mouthpart sensilla in three *Tomicus* beetles.

Sensilla	*T. yunnanensis*		*T. brevipilosus*		*T. minor*	
	Female (µm)	Male (µm)	Female (µm)	Male (µm)	Female (µm)	Male (µm)
Sb.1	46.2 ± 2.3	48.9 ± 3.4	46.3 ± 2.9	50.8 ± 3.3	45.2 ± 3.6	44.4 ± 2.7
Sb.2	4.4 ± 0.2	4.0 ± 0.2	4.0 ± 0.2	4.7 ± 0.2	4.9 ± 0.2	4.4 ± 0.3
Stb.1	3.9 ± 0.1	3.7 ± 0.2	3.8 ± 0.2	3.9 ± 0.1	4.0 ± 0.2	3.9 ± 0.1
Stb.2	2.6 ± 0.2	3.0 ± 0.1	3.5 ± 0.2	3.0 ± 0.1	3.0 ± 0.2	3.4 ± 0.2
Stb.3	1.6 ± 0.1	1.6 ± 0.1	1.6 ± 0.1	1.6 ± 0.1	1.6 ± 0.0	1.5 ± 0.1
Sco	2.3 ± 0.2	2.2 ± 0.2	2.3 ± 0.1	2.1 ± 0.1	2.3 ± 0.2	2.4 ± 0.2
Sch.1	32.1 ± 2.7	32.1 ± 3.0	33.9 ± 3.8	34.6 ± 3.9	33.1 ± 3.6	31.3 ± 3.3
Sch.2	113.2 ±6.6	114.8 ± 7.7	117.0 ± 7.3	117.8 ± 8.3	118.0 ± 6.8	112.9 ± 7.0
Str.1	29.2 ± 2.6	33.1 ± 2.1	32.2 ± 1.3	28.5 ± 1.8	33.2 ± 2.5	29.0 ± 1.8
Str.2	25.6 ± 1.7	24.1 ± 1.5	20.6 ± 1.4	20.4 ± 1.5	15.9 ± 1.5	17.0 ± 0.9
Sdi	11.6 ± 0.4	11.6 ± 0.5	11.2 ± 0.4	11.2 ± 0.6	10.9 ± 0.5	12.8 ± 0.4

**Table 5 insects-14-00933-t005:** The mean numbers of mouthpart sensilla in three *Tomicus* beetles.

Sensilla	*T. yunnanensis*		*T. brevipilosus*		*T. minor*	
	Female	Male	Female	Male	Female	Male
Sb.1	36.1 ± 0.4	36.1 ± 0.4	35.9 ± 0.4	35.9 ± 0.3	36.0 ± 0.2	35.9 ± 0.3
Sb.2	14.1 ± 0.1	14.1 ± 0.1	14.3 ± 0.2	14.3 ± 0.2	14.1 ± 0.1	14.1 ± 0.1
Stb.1	16.1 ± 0.2	16.0 ± 0.2	16.0 ± 0.0	15.9 ± 0.1	16.0 ± 0.0	15.9 ± 0.1
Stb.2	29.9 ± 0.1	29.7 ± 0.2	30.0 ± 0.0	29.9 ± 0.1	29.9 ± 0.1	29.9 ± 0.1
Stb.3	44.1 ± 1.1	44.5 ± 1.2	44.1 ± 1.1	44.5 ± 1.2	44.1 ± 1.1	44.5 ± 1.2
Sco	2.0 ± 0.0	2.0 ± 0.0	2.0 ± 0.0	2.0 ± 0.0	2.0 ± 0.0	2.0 ± 0.0
Sch.1	73.3 ± 2.5	77.3 ± 2.1	87.3 ± 3.5	84.7 ± 2.7	81.1 ± 2.6	83.8 ± 3.2
Sch.2	242.8 ± 2.9	238.3 ± 3.1	221.7 ± 3.3	217.5 ± 2.5	227.9 ± 3.2	226.8 ± 2.3
Str.1	56.9 ± 1.9	57.9 ± 2.1	54.9 ± 1.4	56.0 ± 2.0	57.5 ± 1.6	52.7 ± 1.6
Str.2	200.3 ± 6.5	195.7 ± 6.0	197.9 ± 7.9	198.9 ± 10.0	198.5 ± 8.6	196.5 ± 7.3
Sdi	5.3 ± 0.2	5.1 ± 0.2	5.3 ± 0.2	5.3 ± 0.2	4.9 ± 0.2	5.3 ± 0.2

## Data Availability

The data presented in this study are available in the article.

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
