# Peer review of "Fine Structure of the Mouthparts of Three Tomicus Beetles Co-Infecting Pinus yunnanensis in Southwestern China with Some Functional Comments"

_insects, 2023, doi:10.3390/insects14120933_

Round 1
Reviewer 1 Report
Comments and Suggestions for Authors
The article is broad and detailed. It is well written and I do not see any obvious English mistakes. I only have few comments:
Part 2.1 of the Materials and Methods fits more in the Introduction in my opinion. In this part of the article it might be better to put for example a method of acquiring the insects and the place of collection. The description of bioecological characters, in my opinion, does not fit in this section.
I suggest using terms like „ flexible"/„inflexible" when describing the socket. Those are universally used terms which also help with suggesting a function of the sensillum, since flexible sockets are characteristic of mechanosensilla. These terms are also more descriptive than "tight"/"wide".
For the data availability statement - there were no other photos made that allowed you to reach the conclusions of the research? Usually the dataset in the case of SEM or TEM includes hundreds of photos and only the best of them are included in the article. Especially when you state in the article that you studied 30 specimens only under SEM. Overall data should be stored in the repository and available to the readers.
I suggest accepting the article for publishing after a minor revision.
Author Response
Dear Reviewers of Insects:
Thanks so much for your exceptionally informative revision suggestions to our manuscript (insects-2723253). We accept entirely his/her suggestions. All modifications were labeled with the "Track Changes" function in MS Word. Here, we would outline every response considering the review's questions. The review's questions are in italics in the uploaded file. Please check it.
Reviewer 2 Report
Comments and Suggestions for Authors
Authors described the mouthparts of three bark beetle species, examining different features. The geometroc morphometric section must be modified.
In the Introduction the species were detailed, and the aim of the research was thoroughly discussed.
In the M&M section the landmark configuration should be included, and the issue of the landmarks/semilandmarks should be addressed. According to the deformation grid included in the results, a large number of points were used to define the sghape of the mandibles. However, part of these points should be treated as semilandmarks. Instead, the authors choise to use a software (MorphoJ) in which the semilandmark option is not included. In tpsUtil it is possible to mark which points must be analyzed as semilandmarks, then the tpsRelw software could be used for the PCA and any statistical software for the CVA (using the RWs explaining the 100% of the overall shape variation). An alternative could be the R package GeoMorph of Adams, in which the semilandmaks can be defined and correctly treated. Or also, they could use the aligned configuration in MorphoJ instead of the raw one.
In the Results the sensilla were discussed in details, while the results of the gm analyses should be modified (see above)
If molecular data of these species should be available, the mandible shape variation could be compared to the genetic distance and the phylogenetic signal could be evaluated. This can be done in Morphoj (using the aligned configuration obtained by tpsRelw)
In the discussion the mouthparts features are discussed, but some changes should be done according to the issue of the landmarks/semilandmarks definition.
I suggest the authors to made minor changes in the gm analysis.
Author Response

(The authors gave the same response as above.)

Reviewer 3 Report
Comments and Suggestions for Authors
Authors studied the fine structures of mouthparts and the structures from connected sensilla in three Chinese bark beetles (Tomicus) by scanning electron microscopy (SEM) and transmission electron microscopy (TEM). and compared them with many other coleopteran species. The found minor differences in the mandibles from the four bark beetles and little intraspecific variability in the sensilla. #
Nevertheless, the present findings may help to get more information of the feeding niches of the three Tomerus species as well as on their reproductive behavior.
The study is very carefully done, and the manuscript is professionally written. I found only very few points to be corrected:
- line 65: either use coleopteran (in lowercase letters) or Coleoptera (in uppercase letters)
- line 158: the right and left part of the figure must be changed
- Ref. 23: give book titles either in uppercase or in lowercase letters, but not mixed
Author Response
Dear Reviewers of Insects:
Thanks so much for your exceptionally informative revision suggestions to our manuscript (insects-2723253).
Q1: line 65: either use coleopteran (in lowercase letters) or Coleoptera (in uppercase letters)
Author's response: Okay, we have corrected it in the revised manuscript. Thanks.
Q2: line 158: the right and left part of the figure must be changed.
Author's response: Surely, it is an obvious mistake. We have corrected it. Please check it in the revised manuscript. Thanks a lot.
Q3: Ref. 23: give book titles either in uppercase or in lowercase letters, but not mixed.
Author's response: Here, we used the software EndNote to manage the references to avoid typing mistakes and duplicated references. We also checked the rule of the style that is consistent with the journal Insects. Please check it in the manuscript.